# Australian Communities That Care (CTC) intervention: Benefit-cost analysis of a community-based youth alcohol prevention initiative

Julie Abimanyi-Ochom[1‡]*, Sithara Wanni Arachchige Dona[1‡], Shalika Bohingamu Mudiyanselage[1], Kanika Mehta[2], Margaret Kuklinski[3], Bosco Rowland[4,5], John W. Toumbourou[4], Rob Carter[1‡]

1 Faculty of Health, Deakin Health Economics, School of Health and Social Development, Institute for Health Transformation, Deakin University, Geelong, Victoria, Australia, 2 School of Medicine, IMPACT–The Institute for Mental and Physical Health and Clinical Translation, Deakin University, Geelong, Victoria, Australia, 3 Social Development Research Group, School of Social Work, University of Washington, Seattle, Washington, United States of America, 4 Faculty of Health, School of Psychology, Centre for Social and Early Emotional Development, Deakin University, Geelong, Victoria, Australia, 5 Eastern Health Clinical School & Monash Addiction Research Centre, Monash University, Victoria, Australia

‡ JAO and SWAD are joint first authorship on this work. RC is joint senior authorship on this work.
* j.abimanyiochom@deakin.edu.au

**Data Availability Statement:** All relevant data are within the manuscript and its Supporting Information files.

## Abstract

### Background

Internationally, adolescent alcohol consumption has been a major community concern for decades. Globally, there is a growing array of interventions aimed at preventing youth alcohol-related problems. Notably, the Communities that Care (CTC) process in the USA has proven to be a cost-effective intervention, leading to a reduction in adolescent alcohol-related problems. In Australia, the CTC trial has shown positive outcomes such as reduced adolescent alcohol and drug use, antisocial behaviors, and the availability of alcohol in the community. To encourage wider adoption, it is essential to ascertain the cost-effectiveness of the Australian CTC trial and its potential impact on addressing youth alcohol problems.

### Method

We conducted a limited societal perspective benefit-cost analysis focused on reducing adolescent (aged 10–14) alcohol use in the initial four CTC communities in Australia, spanning 2001 to 2015. To ensure accuracy, estimated benefits were adjusted for joint effects to prevent the double counting of the benefits over time.

### Results

The Australian CTC trial, as an adolescent alcohol prevention strategy, demonstrates robust economic credentials, delivering a return of AUD 2.6 for each dollar invested. The cost savings resulting from the reduction in alcohol consumption were estimated at AUD 123 per youth per 15 years and AUD 8 per person per year. The average cost of the CTC trial

**Funding:** Bosco Rowland, John W Toumbourou and Rob Carter received funding from The National Health and Medical Research Council (APP1087781) and Australian Research Council (LP100200755). The funders had no role in study design, data collection, analysis, decision to publish, or preparation of the manuscript.

**Competing interests:** The following authors served in the not-for-profit company Communities That Care Ltd that was evaluated in this paper: Rowland (current Chief Executive Officer, CEO) and Toumbourou (Chair, Director and ex-CEO). Conflict of interest management strategies were approved by their Deakin University employer. This does not alter our adherence to PLOS ONE policies on sharing data and materials." (as detailed online in our guide for authors http://journals.plos.org/plosone/s/competing-interests).

amounted to AUD 48 per youth over 15 years and AUD 3 per youth per year. The largest contribution to primary benefits were reductions in crime and violence with 93% of the total benefits.

## Conclusions

This study makes a valuable contribution to the international economic evidence related to youth alcohol prevention initiatives. The results affirm the strong cost-effectiveness potential of the initial CTC trial implementation in Australia. Moreover, conducting further cost-benefit modelling for additional outcomes and beyond the intervention's timeframe is likely to enhance the economic viability of the CTC trial.

## Introduction

Globally, excessive alcohol consumption is a leading risk factor for death and disability [1]. Alcohol consumption often begins during the early teenage years, and risky levels of consumption during the teen years are associated with harmful consumption in the adult years [2]. Australia has the highest rates of adolescent alcohol use in the world [3]. If alcohol related health and community issues are to be avoided, cost-effective programs and policies need to be systemically implemented [4]. Australia implemented the Communities That Care (CTC) prevention framework with the aim of reducing and preventing underage alcohol consumption [4]. The CTC framework can also be described as a preventive intervention.

The CTC framework was originally developed in the United States (US) and was later implemented as a community intervention in Australia in 2002 (trial-related prior-implementation activities happened prior to 2002) [5], which aimed at reducing adolescent alcohol and drug use by identifying and addressing local risk and protective factors [6].

CTC is designed to build prevention capacity and sustainability across large municipal communities and is implemented in 5 phases [6–8]. Communities begin by forming a municipal coalition of stakeholders (Phase 1 & 2). They then collect local data and develop a profile of their community (Phase 3) to identify elevated risk factors that cause adolescent alcohol use (e.g., parent and community supply of alcohol, low perceived risk, poor family management). Communities then use this profile to inform the selection of evidence-based strategies that target elevated risk factors (e.g., programs to reduce underage alcohol supply, parent education), while also increasing protective factors such as community rewards and opportunities (Phase 4). Communities then implement and evaluate evidence-based strategies (Phase 5), and monitor their progress by repeating the survey again, and comparing to their Phase 3 results. The CTC trial was conducted in four Australian community coalitions (Local Government Areas (LGA) across Australia (in Victoria and Western Australia) from 2002 to 2015. The evaluation compared 104 non-CTC LGAs [6]. The CTC trial in Australia evaluated data collected over a 15-year period.

The CTC trial action plans focused on reducing family risk factors and favorable attitudes to alcohol and drugs in youth using school-based and parent/family drug education [9, 10]. S1 Table outlines the number of CTC cycles and the duration of the CTC trial in the first four community coalitions. Community coalition 1 completed two cycles compared to the other three communities, which completed only one cycle each.

The CTC trial in US has demonstrated reductions in adolescent alcohol and drug use, and related antisocial behaviors, less favorable parent attitudes toward alcohol and drug use, and

reduced community availability of alcohol [11, 12]. Given these novel findings, this study aims to develop a benefit-cost analysis model to explore the cost and benefits from a limited societal perspective (i.e., all costs and outcomes to whomever they accrue which includes health sector (public/private) or governments or individuals/patients/families, or taxpayer or third party funders [13]) using data from a 15-year evaluation of the Australian CTC intervention in 4 trial communities. Benefit-cost analysis will inform CTC trial effectiveness and guide policy-makers and stakeholders interested in achieving positive youth outcomes in a cost-effective way.

A benefit-cost analysis places a monetary value on significant intervention-related outcomes, based on benefits expected to accrue over the reference period for the participants. The present economic analysis of the Australian CTC trial seeks to determine whether the benefits associated with the observed effects of the CTC trial outweigh the costs of implementation. The benefit-cost analysis results provide economic evidence related to the CTC trial and return on investment of public dollars, even under very conservative cost and benefit assumptions.

## Material and methods

The benefit-cost model estimated the cost of the intervention and the benefits in the youth population based on the reduction in alcohol use due to the CTC trial. Firstly, the reduction in alcohol use measured in the trial was used to estimate the reduction in the alcohol consuming population due to the intervention for the intervention population. Secondly, we estimated the population that have risky alcohol consumption within the alcohol consuming intervention population. Lastly, avoided consequences in health-related events (alcohol-related hospital admissions, emergency department visits, ambulance attendances, alcohol treatment programs), and in non-health events (deaths, and crime) due to risky alcohol consumption were estimated and costed to give the benefit from the CTC intervention.

### Study design

An intervention-based benefit-cost model was conducted from a limited societal perspective.

### Model period

The periods for the benefit-cost model were derived using the Australian Bureau of Statistics (ABS) data periods, which provides five-year population data estimates [14–16]. This provided three model periods covering 15 years of the trial: Period 1; 2001–2005, Period 2; 2006–2010 and Period 3; 2011–2015.

### Model population

The ABS population numbers were aggregated by CTC trial LGAs, age, sex, and year [14–16]. The ABS five-year population numbers were combined with survey estimates of the proportion of alcohol consumption [17] for each age group to estimate the population who do consume alcohol (Fig 1).

### Model inputs

Through interpolation of results of Toumbourou and colleagues study, the Australian CTC trial demonstrated a larger reduction in adolescents using alcohol of 28.3% compared to control communities with a 12.2% reduction over 15 years [6]. The difference in reduction in alcohol use between the two groups was 16.1 percentage points, or an average absolute reduction in use of 1.01% per year (i.e., 1.01 = 16.1/15, 5.05% per five years) [6]. We considered a single

| Period 1: 2001-2005 | 10–14-year-old ABS population x % of population who consumed alcohol | | |
| --- | --- | --- | --- |
| Period 2: 2006-2010 | 10–14-year-old ABS population (new population in Period 2) x % of population who consumed alcohol | 15–19-year-old ABS population (Period 1 population continues to Period 2) x % of population who consumed alcohol | |
| Period 3: 2011-2015 | 10–14-year-old ABS population (new population in Period 3) x % of population who consumed alcohol | 15–19-year-old ABS population (Period 2 population continues to Period 3) x % of population who consumed alcohol | 20–24-year-old ABS population (Period 1 population continues to Period 3) x % of population who consumed alcohol |

Note: x indicates multiplication

**Figure 1 Legend**

| | |
| --- | --- |
| | Cohort started in Period 1 |
| | Cohort started in Period 2 |
| | Cohort started in Period 3 |

**Fig 1. Model population 2001–2015.**

occasion of alcohol risk [18], which is more applicable to the model population. Details of the model inputs are shown in Table 1.

## Model outputs

Model outputs are presented as follows: (1) The Australian CTC trial cost, (2) short-term cost savings (i.e., benefits) from avoided events to health, and criminal system due to CTC trial, and (3) Benefit-Cost ratio (BCR) as primary outcome of CTC trial.

## Data analysis

The cost and benefits estimation methods are presented in Table 2. First, the cost of the CTC trial was calculated for each of the four CTC trial communities using CTC trial records provided by the trial implementation team using a template designed by the health economics investigators (See S1 Fig). CTC costs were attributed to organizing the community coalitions,

**Table 1. Benefit-cost model inputs.**

| Model input | Model Value | Reference and notes |
|---|---|---|
| Average absolute reduction in alcohol use due to CTC trial | 1.01% per year | Toumbourou et al., 2019 [6] |
| Percentage of adolescents that use alcohol (10–17 years old) | 61.8% | Kelly et al., 2015 [17] |
| Percentage of adolescents within the intervention community exposed to the CTC trial | 100% | Kuklinski et al., 2012 [19] |
| **Alcohol risk** Percentage of youth using alcohol at levels that risk harm: 2006–2010 | 18–19 years old Males: 18.5%, Females: 15.9% | 2007 AIHW estimated alcohol-related harm from a single occasion (more than 4 drinks) by age and sex (17.2% gender aggregated rate) [20] |
| **Alcohol risk** Percentage of youth using alcohol at levels that risk harm: 2011–2015 | 18–19 years old Males: 13.1%, Females: 15.8% 20–24 years old Males: 13.1%, Females: 15.8% | 2013 AIHW alcohol-related harm from a single occasion (more than 4 drinks) by age and sex (gender aggregated rate: 18–19 years old—8.3%, 20–24 years old–14.3%) [21] |
| Percentage of youth alcohol-related emergency department visits and cost per visit | 2001, 2006, 2011: 15.1%, 24.4%, 28% $290 (2005) | 15–24 age rate used, cost [22, 23] |
| Percentage of youth alcohol-related hospital admissions and cost per admission | 2001, 2006, 2011: 29.3%, 39.9%, 47.6% $22,375 (2005) | 15–24 age rate used, cost [22, 23] |
| Percentage of youth alcohol related ambulance attendance and cost per attendance | 2001, 2006, 2011: 17.9%,17.9%,56.2% $689 (2012) | 15–24 age rate used, cost [22, 23] |
| Percentage of youth alcohol related alcohol treatment programs and cost per program, | 2001, 2006, 2011: 45.4%, 60.5%, 50.3% $7,285 (2011) | 15–24 age rate used, cost [23, 25] |
| Percentage of youth alcohol-related deaths and cost per life lost | 2001, 2006, 2011: 0.67%, 0.67%, 0.52% $1,365,563 (2010) | 15–24 age rate used, cost [23, 26] |
| Number of overall crimes (age 10–19) | 109,844 (age 10–19) | Offenders (2008–09) [27] |
| Percentage of offenders involved in alcohol-related incidents | 1.4% (age 10–14), 18.8 (age 15–19) and 25.4% (age 20–24) | 2007 [28] |
| Alcohol attributable fraction of crime | 15% | in 2008, 2010 & 2015 [29] |

Note: AIHW- Australian Institute of Health and Welfare

intervention programs, student surveys, rental space, salary, CTC training for each of the 5 phases and other costs such as transport, catering resources, and report development. Activity costs (categories) were estimated using an approach [26] of multiplying the quantity of each resource by its unit price to generate a resource cost. The costs of all resources used in an activity were then summed to give the total cost of the activity. The activity costs were summed up to give the total cost per community for the intervention period. Only the first cycle was considered when there were multiple cycles during the intervention period for comparability across the four communities (see S1 Fig) (three of the four communities had only one cycle).

Second, societal benefits from averted events resulting from the reduced alcohol consumption due to the CTC trial were modelled and calculated for the 15-year intervention period [30]. The model assumption was that the reduction in alcohol consumption due to the CTC trial [6] led to the reduction in alcohol-related adverse events and associated costs during the intervention period. This assumption was derived from the established evidence between risky alcohol consumption and alcohol related adverse events based on the National Health and

**Table 2. Cost and benefits estimation methods.**

| Costs and benefits | Measures | Estimations and assumptions | Data source | Cost and benefits estimation method | Cost and benefits estimations |
|---|---|---|---|---|---|
| **Intervention cost** | Total intervention implementation cost | Expenditure *Categories*<br>1. Administration cost (organizing the community coalitions intervention programs, student surveys)<br>2. Rental space cost<br>3. Staff salary<br>4. CTC training for each of the five phases<br>5. Transport<br>6. Catering resources<br>7. Report development | CTC implementation team | CTC trial cost for each expenditure category per ABS youth aged 10–14 years who lived in the study communities in 2000–2015 | Total CTC trial cost for each expenditure category. The average cost per community weighted by community size (weighted average), which reduces the impact of population size on average cost for small population communities. |
| **Benefits of the intervention** | | | | | |
| <u>Avoided health system events</u> | | | | | |
| **alcohol-related Emergency Department (ED) presentations** | 1. Rate of avoided youth alcohol cases, 5.05% for five years | | Toumbourou et al., 2019 [6] | The total population in the CTC communities x 5.05% | Total number of avoided youth alcohol cases in the CTC communities |
| | 2. Percentage of alcohol-attributed injuries at EDs, 15.1%, 24.4% & 28.0% | The proportion of alcohol-attributed injuries presented at EDs in 2001, 2006 & 2011 | Turning point [23] | (Total number of avoided youth cases in the CTC communities (5.05%/5-years) x the proportion of alcohol-attributed injuries presented at EDs per 10,000 population) | Total number of avoided alcohol-attributed ED presentations |
| | 2. Cost per ED presentation, $ 254 | Average ED treatment costs for all states and territories, considering both acute and non-acute ED expenses | Turning point [23] | Total number of avoided alcohol-attributed ED presentations x $254 | The total cost saved from avoided alcohol-attributed ED presentations |
| **alcohol-related hospital admissions** | 1. Rate of avoided youth alcohol cases, 5.05% for five years | | Toumbourou et al., 2019 [6, 17] | The total population in the CTC communities x 5.05% | Total number of avoided youth cases in the CTC communities |
| | 2. Percentage of alcohol-attributed hospital admissions, 29.3%, 39.9% & 47.6% | The proportion of alcohol-attributed hospital admissions Eds in 2001, 2006 & 2011 | Turning point [23] | (Total number of avoided youth cases in the CTC communities x the proportion of alcohol-attributed hospital admissions per 10,000 population) | Total number of avoided alcohol-attributed hospital admissions |
| | 2. Cost per hospital admission, $ 22,357 | Average hospital admission treatment cost | Laslett et al., 2010 [22] | Total number of avoided alcohol-attributed hospital admissions x $22,357 | The total cost saved from avoided alcohol-attributed hospital admissions |
| **alcohol-related ambulance attendances** | 1. Rate of avoided youth alcohol cases, 5.05% for five years | | Toumbourou et al., 2019 [6, 17] | The total population in the CTC communities x 5.05% | Total number of avoided youth cases in the CTC communities |
| | 2. Percentage of alcohol-attributed ambulance attendance, 17.9%, 17.9% & 56.2% | The proportion of alcohol-attributed ambulance attendance in 2001, 2006 & 2011 | Turning point [23] | (Total number of avoided youth cases in the CTC communities x the proportion of alcohol-attributed ambulance attendance per 10,000 population) | Total number of avoided alcohol-attributed ambulance attendance |
| | 2. Cost per ambulance attendance, $ 689 | Average ambulance attendance cost | Watts et al., 2013 [24] | Total number of avoided alcohol-attributed hospital admissions x $689 | The total cost saved from avoided alcohol-attributed ambulance attendance |
| **alcohol treatment programs** | 1. Rate of avoided youth alcohol cases, 5.05% for five years | | Toumbourou et al., 2019 [6] | The total population in the CTC communities x 5.05% | Total number of avoided youth cases in the CTC communities |

(*Continued*)

**Table 2.** (Continued)

| Costs and benefits | Measures | Estimations and assumptions | Data source | Cost and benefits estimation method | Cost and benefits estimations |
|---|---|---|---|---|---|
| | 2. Attendance percentage for alcohol-attributed treatment programs 45.4%, 60.5% & 50.3% | Participation proportion for alcohol-attributed treatment programs in 2001, 2006 & 2011 | Turning point [23] | (Total number of avoided youth cases in the CTC communities x the participation proportion for alcohol-attributed treatment programs per 10,000 population) | Total number of avoided participations in alcohol-attributed treatment programs |
| | 1. Treatment cost per case, $7285 | The average treatment cost per case: residential drug treatment, non-residential treatment and pharmacotherapeutic treatments | Smith et al., 2014 [25] | Total number of avoided participations in alcohol-attributed treatment programs x $7285 | The total cost saved from avoided participants in alcohol-attributed treatment programs. |
| Avoided non-health system events | | | | | |
| **overall crimes** | 1. Rate of avoided alcohol consumption, 5.05% for five years | | Toumbourou et al., 2019 [6] | The total population in the CTC communities x 5.05% | Total number of avoided youth cases in the CTC communities |
| | 2. % of offenders by age involved in alcohol-attributed incidents | | Palk et al., 2007 [28] | Total number of avoided youth cases in the CTC communities x % of offenders by age | Total number of avoided crimes |
| | 3. Alcohol attributable crime cost per person, $28,603.66 | Alcohol attributable crime cost in 2004–05 | Collins et al., 2008 [29] | Alcohol attributable crime cost in 2004–05 x total number of avoided crimes | The total cost saved from avoided crimes |
| **alcohol-related deaths** | 1. Rate of avoided alcohol consumption, 5.05% for five years | | Toumbourou et al., 2019 [6] | The total population in the CTC communities x 5.05% | Total number of avoided youth cases in the CTC communities |
| | 2. Percentage of alcohol-attributed deaths, 0.67%, 0.67% & 0.52% | The proportion of alcohol-attributed deaths in 2001, 2006 & 2011 | Turning point [23] | (Total number of avoided youth cases in the CTC communities x the proportion of alcohol-attributed deaths per 10,000 population) | Total number of avoided alcohol-attributed deaths |
| | 2. Cost per death, $1,367,230 | Average cost per incident (reduced workforce and reduced household labor from premature mortality) | Manning et al., 2013 [26] | Total number of avoided alcohol-attributed deaths x $1,367,230 | The total cost saved from avoided alcohol-attributed deaths |

Medical Research Council (NHMRC) guidelines. The reduction in alcohol-related adverse events were estimated for risky alcohol consumption based on the NHMRC guidelines [31].

The Australian low-risk drinking guidelines recommend that adolescents should not consume alcohol before the age of 18 [31]. Thus, the model assumed that for the 10 to 17-year-old population, any alcohol consumed is harmful. For the 18 to 19-year-olds and 20 to 24-year-olds alcohol was assumed harmful when consumed above the NHMRC recommended levels, more than four standard drinks in a session.

Avoided adverse events for alcohol users were categorised as health and non-health related. As shown in Table 2, the health-related avoided events include alcohol-related: 1) hospital admissions, 2) emergency department (ED) visits, 3) ambulance attendances, 4) alcohol treatment programs. The non-health events include alcohol-related: (1) deaths, and 2) all types of crimes.

The benefits were adjusted for joint effects using a base model of 50%, where the total benefits were halved, based on previous literature [32] adjusted for possible double counting due to correlations among benefits from different outcomes (e.g., when the same patient utilises the

ambulance, ED, and in-hospital care for a hospital admission, these events were considered as three separate patients and then the join effect was applied to attribute those events to that one patient) [33]. The joint effect was considered to correct for shared risk factors that can contribute to the same outcomes. All nominal costs and benefits estimates were converted to 2020 Australian dollars (AUD) using the health price index [34, 35]. The model was generated using Microsoft Excel. The impact of intangible costs and benefits such as stress and quality of life were not included in the model. The total, average, weighted average intervention cost, total and average benefits, and average dollars saved per youth were calculated. The benefit-cost ratio was calculated as the ratio of the average dollar saved per youth and average cost per youth.

### Sensitivity analysis

A sensitivity analysis was undertaken to deal with uncertainty in average absolute reduction due to the CTC trial to ensure the reliability of the BCR results and conclusions. Bootstrapping was used for sensitivity analysis for the model variables. The 'average absolute reduction' of 1.01% per annum was modelled to account for fading intervention effects with ± 5 and 10% variation. The variation of the proportion of youths exposed to the intervention was ± 5 and 10% of the base model. Similarly, the variation of other benefits variables was also applied through the sensitivity analysis.

## Results

The total youth population aged 10–14 in the four community coalitions assumed to be impacted during the intervention period from 2001 to 2015 was 48,210.

### CTC costs

The total cost of implementing the Australia CTC trial for the four pioneer community coalitions over 15 years was estimated at AUD 2.3 million (Table 3). The majority of total

**Table 3. CTC trial costs by cost category (AUD2020).**

| | Duration of CTC (Years) | Cost categories AUD2020 | | | | | | | Population | Total CTC trial cost | Total cost per youth | Total cost per year per youth |
|---|---|---|---|---|---|---|---|---|---|---|---|---|
| | | Community mobilization cost | CTC trial program cost | Student survey cost | Rental space cost | Salary allocation costs | CTC training cost | Other costs including transport | | | | |
| Community coalition 1 | 6 | 173,892 | 17,602 | 49,724 | 7,661 | 201,755 | 1574,33 | - | 28,003 | 608,067 | **8.74** | **0.58** |
| Community coalition 2 | 13 | 121,432 | 15,626 | 88,249 | 8,616 | 280,826 | 186,802 | - | 12,199 | **704,799** | 12.61 | 0.79 |
| Community coalition 3 | 10 | 89,387 | 15,626 | 65,058 | 5,607 | 270,593 | 127,265 | 5,949 | 4,409 | 579,484 | **14.62** | 0.79 |
| Community coalition 4 | 7 | 46,066 | 36,096 | 27,153 | 7,421 | 220,770 | 83,680 | - | 3,599 | **421,187** | 12.02 | **1.12** |
| **Total** | | 430,777 | 84,950 | 230,184 | 29,305 | 973,944 | 555,180 | 5,949 | 48,210 | 2,313,537 | 48 | 3 |
| **Proportion of total CTC trial cost** | | 19% | 4% | 10% | 1% | 42% | 24% | 0.3% | - | - | - | - |

Note: Costs are reported only for the first cycle when there are two cycles (community coalition 1). Bold figures represent the highest and lowest costs. The intervention costs were provided by the CTC trial team (based on actual expenses during the trial period, (considering the unit costs of each cost category for each community coalition)). The intervention costs accounted for all the costs relevant to delivering the intervention including on-costs.

**Table 4. CTC trial benefits (avoided costs) (AUD2020) (adjusted for joint effects).**

| CTC trial benefits (avoided costs due to alcohol consumption reduction) | Total benefit (AUD) | Benefit percentage of total benefits | Total benefit per youth for the intervention period (AUD) | Benefit per youth per year (AUD) |
|---|---|---|---|---|
| **Health-related benefits** | 281,470 | 4.8% | 5.8 | <1 |
| Alcohol-related hospital admissions | 204,747 | 3.5% | 4.2 | <1 |
| Alcohol-related ED visits | 1,360 | <1% | <1 | <1 |
| Alcohol-related ambulance attendance | 3,905 | <1% | <1 | <1 |
| Alcohol-related treatment programs | 71,458 | 1.2% | 1.5 | <1 |
| **Non-health related benefits** | 5,638,549 | 95.3% | 117 | 7.8 |
| Alcohol-related deaths (Mortality) | 157,767 | 2.7% | 3.3 | <1 |
| Alcohol-related all crimes | 5,480,782 | 92.6% | 113.7 | 7.6 |
| **Total benefits** | 5,920,018 | 100% | 123 | 8 |

intervention costs were attributed to staff salary allocations (42%). Considering only the first cycle for all included community coalitions, community coalition 2 had the highest intervention cost estimated at AUD 0.70 million, and community coalition 4 had the lowest intervention cost, estimated at AUD 0.42 million. The CTC trial cost per youth was AUD 48 for the entire intervention period and AUD 3 per year per youth.

Regarding the CTC trial cost per youth, community coalition 1 had the lowest average intervention cost, with AUD 8.74 per youth, while community coalition 3 had the highest cost, at AUD 14.62 per youth. In terms of the cost per youth per year for the CTC trial, community coalition 1 had the lowest average intervention cost, at AUD 0.58 per youth per year, while community coalition 4 had the highest annual cost, at AUD 1.25 per youth.

## CTC overall benefits

The avoided cost from alcohol-related events resulting from the CTC trial was estimated at AUD 5.9 million (as detailed in Table 4) over the 15-year period from 2001–2015. Notably, the highest avoided cost was attributed to alcohol-related crimes at AUD 5.5 million (93%), while the lowest was associated with ED visits, totaling AUD 1,360 (< 1%). On average, the benefit (avoided cost), per youth was AUD 123 for the entire intervention period and AUD 8 per year.

Ninety-five percent of benefits were from non-health related avoided costs, while the other 5% were from health-related avoided costs. According to the estimated total cost and total benefits, the benefit cost ratio is 2.6, which translates to a return of AUD 2.6 for each dollar invested.

## Sensitivity analysis

The sensitivity analysis found the model outputs to be robust. The BCR marginally differed for each variable with ± 5 and ±10 variations. The lowest BCR was estimated at 2.34 when the average absolute reduction was assumed to -10% of 1.01, while the highest BCR was at 2.86 when the average absolute reduction was assumed to +10% of 1.01.

## Discussion

This study reports the benefits and cost of the inaugural Australian CTC trial undertaken with 4 CTC trial communities, revealing that each dollar spent on the CTC trial per youth per year resulted in an estimated savings of AUD 2.6, primarily attributed to avoided alcohol-related crime events. These findings offer valuable insights into quantifying and monetizing the benefits of community-wide adolescent alcohol prevention initiatives within the Australian context.

They underscore the potential of the CTC prevention framework to improve adolescent health and behavioral outcomes in Australia.

Intervention costs are lower in Australia, compared to the US [19]. The Australian CTC trial cost was AUD 3 per youth per year compared to AUD 199 per youth per year for the US CTC trial (2004 USD 513 per youth for 5 years/ 102 per youth per year). The main reason for a lower cost in the Australia CTC was that community coalition service delivery budgets were redeployed in Australia compared to the US where new budgets were introduced for the CTC trials. In Australia there was also a greater use of community alcohol supply reduction interventions (e.g., underage test purchasing, retailer warning letters and media releases) which were less costly than the family and school interventions used in the US. It is not entirely comparable to contrast the US return on investment with the finding of return of investment for this Australian study. T. The return for the US CTC trial was AUD 10.3 (2004 USD 5.3) for each dollar spent compared to a return of AUD 2.6 for each dollar spent in the Australian CTC trial [19]. However, the US analysis included both the long (lifetime) and short term (limiting to intervention period) harms prevented from tobacco use, delinquency, and school dropout [36].

The Australian CTC trial demonstrated significant prevention effects on adolescent lifetime alcohol, tobacco, and cannabis use and past year antisocial behavior [10]. However, the current BCR analysis was conservative as it only included impacts associated with reduced alcohol consumption over the intervention period of 15 years. Importantly, this analysis demonstrates that preventive interventions can incur significant savings related to youth alcohol use over 15 years. Other Australian studies examining the benefit-cost of reducing alcohol consumption from a societal perspective have identified larger life-time cost savings as alcohol causes high costs to the society [29, 36]. However, these studies have examined adult populations and included both long- and short-term economic impacts at the population level [29]. Given that our study only examined benefits during the intervention period using community level data, modelling lifetime benefits may identify greater benefit-cost savings for Australian communities, as evidenced from the US CTC evaluation [19].

The major component of costs for implementing CTC in this trial was related to staff and training. This training focused on developing stakeholder knowledge of prevention science and teaching them how to collect and critically interpret appropriate prevention data. The data helped them strategically and systematically prioritize behavioral outcomes and recognize and implement tested effective programs [37]. As the CTC trial is also designed to build community capacity, the costs needed to implement the CTC trial will likely diminish over time. In the subsequent CTC cycles, communities will require less training to progress through the CTC phases or will be able to facilitate their own training with minimal support. As an example, in the second cycle for the community coalition community with two cycles, the cost was reduced by AUD 65,736 (displaying probable economies of scale as the number of CTC cycles increase).

Most community coalition alcohol prevention programs were funded by redeploying or reallocation of existing resources. Therefore, the opportunity costs of the Australian CTC trial implementation were lower compared to the few communities that implemented utterly new programs. This was particularly important for the feasibility of the CTC trial implementation in small communities with smaller populations and fewer resources (the sample ranged from 3,599 to 28,003 in the four community coalition communities). This suggests that return on investment for implementing the CTC framework can be partly linked with assisting community investment to reorient health and evidence-based prevention services to meet community needs more efficiently, rather than implementing and developing entirely new services. It suggests that mobilizing communities to work within a prevention framework to implement evidence-based programs or reorient services can be cost-beneficial. The implementation of CTC

in Australia is mainly through the re-allocation of existing resources, which should help with the sustainability of CTC over time. A call for communities to reorient health services has been advocated since the launch of the Ottawa Charter for Health Promotion, in 1986 [38].

Other community prevention interventions, which used experimental designs, have also demonstrated efficacy in preventing adolescent alcohol use [39]. Like CTC, these interventions depended on mobilizing communities and long-term prevention investment. Thus, the findings of our study suggest that further examination of the economic benefits of CTC using an experimental design would be a worthy investment. If possible, it would be essential to compare the average absolute reduction of the BCR for different community approaches and identify whether a particular approach is more effective and efficient. Based on the model results, the CTC trial had a positive impact on crime due to reduction in alcohol consumption among youths and therefore continuation of these kinds of interventions could significantly reduce crime.

A strength of the study is its naturalistic evaluation context and, therefore, its high ecological validity. It used data collected over a 15-year period to evaluate trends that are influenced by real life factors and community demands. Moreover, the sensitivity analysis demonstrated the results are robust with marginal changes in BCR, with the Australian CTC trial still worthwhile (cost beneficial overall) after adjustment of the different rates used in the benefits model.

## Limitations and implications

The Australian CTC trial study was conducted as a pragmatic community coalition trial, with 4 trial communities, and these communities were not randomized. Data sources used did not account for differences in rates of alcohol consumption and alcohol-related events by state (e.g. Victoria and Western Australia) [40]. Thus, the model may have underestimated CTC trial benefits for Western Australian communities, since alcohol consumption is generally higher in these communities [41].

Self-reported data on alcohol use is a limitation due to the potential for response bias. Since adolescents are generally not expected to consume alcohol, these self-reported measures may under-report actual consumption levels, thereby obscuring the true benefits of the intervention. Alcohol attributable event rates by age and year of occurrence were used to estimate benefits from reduced ED, hospital admission, ambulance attendance, treatment programs, and death. It is possible that using the same rate by gender may have overestimated benefits for females and underestimated benefits for males. Furthermore, rates on ambulance attendance used 2006/07 rates for missing 2001 values, and rates on death used 2001 rates for missing 2006 rates, which may have underestimated and overestimated benefits respectively.

The model assumes that a reduction in alcohol consumption will lead to proportional decreases in adverse events. However, this relationship is not always linear, and reductions in alcohol use do not consistently result in equivalent decreases in adverse events.

Alcohol consumption rates vary by gender and age, but this was not accounted for based on data available on the CTC trial average absolute reductions in the reference sources [6]. Using an average absolute reduction may have overestimated or underestimated the average absolute reduction for females or males, respectively. The annualized rate of the average absolute reduction over the intervention period does not account for the fading effect of the intervention. However, sensitivity analysis explored uncertainty around the average absolute reduction, and the BCR was still greater than 1, hence even with a possible fading intervention effect, the CTC trial was still judged to be beneficial.

Similarly, the alcohol attribution fraction of crime was consistent across the period, which probably led to underestimating the benefits. The measured police-reported crime prevention

benefit in the relevant Australian CTC evaluation [42] is consistent with that attributed in the current analysis. The national rates of alcohol-related harm may not be specific to the CTC communities. Rates associated with alcohol-related events may include harm due to youth alcohol use and harms to youths due to other alcohol users. The model adjusted for joint effects, by multiplying benefits by a factor of 0.5 using [32]. However, there is limited evidence on alternative methods to adjust for joint effects. The model input- rates for our study were based on the literature (except for alcohol reduction rates). Various benefits typically included in similar analyses, especially for long term or lifetime impacts, were not included in this analysis. This included factors such as effects on school dropouts, employment and intangible benefits including quality of life and wellbeing. Therefore, future research on the Australian CTC should aim to estimate long term or lifetime benefits by incorporating these benefits.

## Conclusion

The benefit-cost ratio of the Australian CTC trial was found to be 2.6, which demonstrates that benefits (avoided alcohol-related costs) outweighed the Australian CTC trial cost across 15 years. The study suggests that the CTC trial implementation is worthwhile. Findings also suggest that CTC offers a cost-beneficial evidence-based approach for use by coalition communities, governments, and large funding bodies to bring about population change in alcohol-related health and social outcomes. This paper extends the range of cost-benefit interventions available in the international context to prevent adolescent health and social problems. Future research should examine the economic benefits of prevented youth tobacco and cannabis use as well as antisocial behavior as demonstrated in the Australian CTC Trial [6]. Additionally, it should utilise more robust experimental designs, including randomised control trials, and consider average absolute reductions that accounts for age, gender, length of intervention period, and long-term benefits for cost effectiveness analysis.

## Supporting information

**S1 Table. Duration of the Australian CTC implementation.**
(DOCX)

**S1 Fig. Example anonymised Australian CTC community cost template.**
(DOCX)

## Author Contributions

**Conceptualization:** Julie Abimanyi-Ochom, Bosco Rowland, John W. Toumbourou, Rob Carter.

**Data curation:** Julie Abimanyi-Ochom, Sithara Wanni Arachchige Dona, John W. Toumbourou.

**Formal analysis:** Julie Abimanyi-Ochom, Sithara Wanni Arachchige Dona, Shalika Bohingamu Mudiyanselage, Kanika Mehta.

**Funding acquisition:** Bosco Rowland, John W. Toumbourou, Rob Carter.

**Investigation:** Julie Abimanyi-Ochom.

**Methodology:** Julie Abimanyi-Ochom, Sithara Wanni Arachchige Dona, Shalika Bohingamu Mudiyanselage, Rob Carter.

**Supervision:** Julie Abimanyi-Ochom, Rob Carter.

**Validation:** Margaret Kuklinski.

**Writing – original draft:** Julie Abimanyi-Ochom, Sithara Wanni Arachchige Dona.

**Writing – review & editing:** Julie Abimanyi-Ochom, Sithara Wanni Arachchige Dona, Shalika Bohingamu Mudiyanselage, Margaret Kuklinski, Bosco Rowland, John W. Toumbourou, Rob Carter.

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
