## [Decision Letter · Decision Letter 0]

11 Jun 2024

PONE-D-24-08327Australian Communities That Care (CTC) intervention: Benefit-cost analysis of a community-based youth alcohol prevention initiativePLOS ONE

Dear Dr. Abimanyi-Ochom,

Thank you for submitting your manuscript to PLOS ONE. After careful consideration, we feel that it has merit but does not fully meet PLOS ONE’s publication criteria as it currently stands. Therefore, we invite you to submit a revised version of the manuscript that addresses the points raised during the review process.

We look forward to receiving your revised manuscript.

Kind regards,

Daryl Higgins, PhD

Academic Editor

PLOS ONE

Journal Requirements:

 [Bosco Rowland, John W Toumbourou and Rob Carter received funding from The National Health and Medical Research Council (APP1087781) and Australian Research Council (LP100200755).].  

[Bosco Rowland, John W Toumbourou and Rob Carter received funding from the National Health and Medical Research Council (APP1087781) and Australian Research Council (LP100200755).]

  [Bosco Rowland, John W Toumbourou and Rob Carter received funding from The National Health and Medical Research Council (APP1087781) and Australian Research Council (LP100200755).].  

[The following authors served in the not-for-profit company Communities That Care Ltd that was evaluated in this paper:

Rowland (current Chief Executive Officer, CEO) and Toumbourou (Director and ex-CEO). Conflict of interest management strategies were approved by their Deakin University employer.]. 

5. We note that you have indicated that there are restrictions to data sharing for this study. PLOS only allows data to be available upon request if there are legal or ethical restrictions on sharing data publicly. For more information on unacceptable data access restrictions, please see http://journals.plos.org/plosone/s/data-availability#loc-unacceptable-data-access-restrictions. 

6. In the online submission form, you indicated that [The benefit-cost model data includes multiple excel files with spreadsheets and are stored in the University secured data storage. Data can be provided on request from authors.]. 

Additional Editor Comments:

Despite a significant number of attempts, I haven't been able to secure 2 reviews. However, I agree with the review comments that I have received, and so as to not hold up the process any further, I am making my recommendation on the basis of the review that I have received.

Reviewers' comments:

Reviewer's Responses to Questions

**Comments to the Author**

1. Is the manuscript technically sound, and do the data support the conclusions?

Reviewer #1: No

2. Has the statistical analysis been performed appropriately and rigorously? 

Reviewer #1: I Don't Know

3. Have the authors made all data underlying the findings in their manuscript fully available?

Reviewer #1: Yes

4. Is the manuscript presented in an intelligible fashion and written in standard English?

Reviewer #1: Yes

5. Review Comments to the Author

Reviewer #1: Australian Communities that Care Intervention: B-C analysis of a community-based youth alcohol prevention initiative.

As the authors note adolescent and youth alcohol drinking is a public health concern. And evidence on what works and is cost-effective is valuable.

However despite being nicely written the underlying logic is not convincing, and in particular whether the trial outcomes that inform this B-C analysis are suitable to answer the research question.

A Cost - Benefit analysis relies entirely on the interpretation of reported outcomes, in this case of a CTC intervention in 5 localities. As such it would be useful to restate the key outcomes being draw on and what they mean. As I understand the outcomes from the Toumbourou et al 2019 paper as reported, the key question related to alcohol assumption was about number of drinks a year 8 student had even consumed - defined as more than a few sips of an alcoholic beverage ever – dichotomised into ever vs never. It is not self-evident that this would be a useful descriptor of alcohol consumption from which to model impacts of alcohol consumption on ED attendance, criminal behaviour etc – which might rather have been related more to intoxication. If this is a mis-interpretation please describe more clearly the trial data on which the modelling is based. But if correct, greater attention as to why this outcome is considered sound for assessing the impacts of alcohol needs to be made. The B-C analysis rests on this.

In particular, given the vast majority of benefits were from a modelled crime reduction, evidence to support the plausibility of the assumptions here are crucial. For example does a PAF for alcohol related youth crime, necessarily translate into what would be avoided if alcohol consumption was reduced – is the PAF is based on an association or a causal link. Further in terms of evidence from this study, perhaps explain why a 1%pa reduced likely of ever having a drink in year 8 in a community is linked to the number of adolescents who might get intoxicated and perhaps more likely to participate in crime. There seems to be a few logic steps missing.

The survey on which the CTC evaluation was based did include questions on anti-social behaviour including involvement in criminal activity – but it appears this wasn’t used in the C-B analysis. It is not clear why not?

Finally, ensuring the intervention is fully costed is clearly important. I could not locate the details on how the costing was done. Specifically what was the source of the costing data, do costs reflect a full cost,for example if office space were made available at below-market rate – how was that treated? It is unclear what is covered under salary costs – does it include wage on-costs and salary overheads (eg for HR, financial services, IT other agency costs).

6. PLOS authors have the option to publish the peer review history of their article (what does this mean?). If published, this will include your full peer review and any attached files.

Reviewer #1: No

---

## [Author Response · Author response to Decision Letter 0]

1 Aug 2024

The response to the specific reviewer and editor comments have been included in the response to reviewers document.

---

## [Decision Letter · Decision Letter 1]

1 Oct 2024

PONE-D-24-08327R1Australian Communities That Care (CTC) intervention: Benefit-cost analysis of a community-based youth alcohol prevention initiativePLOS ONE

Dear Dr. Abimanyi-Ochom,

Thank you for submitting your manuscript to PLOS ONE. After careful consideration, we feel that it has merit but does not fully meet PLOS ONE’s publication criteria as it currently stands. Therefore, we invite you to submit a revised version of the manuscript that addresses the points raised during the review process.

We look forward to receiving your revised manuscript.

Kind regards,

Daryl Higgins, PhD

Academic Editor

PLOS ONE

Journal Requirements:

Reviewers' comments:

Reviewer's Responses to Questions

**Comments to the Author**

1. If the authors have adequately addressed your comments raised in a previous round of review and you feel that this manuscript is now acceptable for publication, you may indicate that here to bypass the “Comments to the Author” section, enter your conflict of interest statement in the “Confidential to Editor” section, and submit your "Accept" recommendation.

Reviewer #2: All comments have been addressed

2. Is the manuscript technically sound, and do the data support the conclusions?

Reviewer #2: Partly

3. Has the statistical analysis been performed appropriately and rigorously? 

Reviewer #2: Yes

4. Have the authors made all data underlying the findings in their manuscript fully available?

Reviewer #2: Yes

5. Is the manuscript presented in an intelligible fashion and written in standard English?

Reviewer #2: Yes

6. Review Comments to the Author

Reviewer #2: The key areas for improvement are suggested below, I have attached a more detailed feedback in the track-changed Word document.

1. Assumptions: Oversimplification of the relationship between alcohol use and health outcomes (e.g., proportional reductions in adverse events). The assumption of confounding variables not being controlled needs to be addressed (e.g., socioeconomic status, mental health conditions).

2. Methodology: Limiting analysis to the first cycle of the intervention may underestimate long-term costs. Clarification on long-term benefits in the discussion is required. Confounding factors like varying levels of alcohol reduction are not accounted for (e.g., small vs large reductions in consumption).

3.Definitions: Expanding on definitions (e.g., examples of crimes attributed to alcohol use) to aid non-expert readers.

4.Future Recommendations: Suggestion to include a cost-effectiveness analysis in future research. Highlighting the limitations of self-reported data (e.g., potential response bias).

7. PLOS authors have the option to publish the peer review history of their article (what does this mean?). If published, this will include your full peer review and any attached files.

Reviewer #2: **Yes: **Dhatsayini Rattambige

---

## [Author Response · Author response to Decision Letter 1]

23 Oct 2024

Dear Reviewers, 

Thank you for the opportunity to revise our manuscript. We appreciate the valuable feedback provided. Please see attached document "Response to Reviewer Comments" for our response to all the reviewer comments (also copied below).

Response to reviewers 

# 

Reviewer comment 

Response 

1 

While these estimates are drawn from established sources, the analysis assumes that any reduction in alcohol use will directly lead to proportional reductions in adverse events. This may oversimplify the relationship between alcohol use and health outcomes, as reductions in use do not always translate linearly to avoided event - would you be able to highlight this in the discussion/limitation section? 

We appreciate this comment. As suggested, this limitation was included in the discussion section. 

The change can be found on Page 23, line 369- 371 

The model assumes that a reduction in alcohol consumption will lead to proportional decreases in adverse events. However, this relationship is not always linear, and reductions in alcohol use do not consistently result in equivalent decreases in adverse events. 

2 

The analysis only considers the costs from the first cycle of the intervention, despite acknowledging that there were multiple cycles during the intervention period limiting the analysis to the first cycle, the total long-term costs of maintaining or repeating the intervention are not fully captured. It may potentially underestimate the true costs of scaling or sustaining the intervention over time - however in the discussion long-term benefits are discussed in the discussion, could you clarify this? 

Thank you for this comment. To clarify this, three of the four communities had one cycle; therefore, the cost was only associated with one cycle for comparability across the four communities. However, future CTCs can consider all the cycles. The evidence from this study shows that the cost per youth/year reduces over future cycles, highlighting possible economies of scale as the number of CTCs cycles increase. 

We have revised this on Page 9, line 176 -179 

Only the first cycle was considered when there were multiple cycles during the intervention period for comparability across the four communities (see S2 Example anonymised Australian CTC community cost template) (three of the four communities had only one cycle). 

Also revised on page 21, Line 317-318 

 (displaying probable economies of scale as the number of CTCs cycles increase). 

3 

how was this assumption established? were any other variables controlled? - If you have the data to control any confounding variables, perhaps add it to the analysis, if not it may need to be acknowledged in the discussions 

We appreciate this comment. This assumption was derived from the established evidence between risky alcohol consumption and alcohol related adverse events based on the NHMRC. Australian guidelines to reduce health risks from drinking alcohol. Canberra, ACT: National Health and Medical Research Council; 2009 

We have specified this on page 9, line 184 to 187 

This assumption was derived from the established evidence between risky alcohol consumption and alcohol related adverse events based on the National Health and Medical Research Council (NHMRC) guidelines. The reduction in alcohol-related adverse events were estimated for risky alcohol consumption based on the NHMRC guidelines 31. 

4 

Would it be beneficial to expand these definitions by providing examples of what these measure? for example I would be interested to understand what crimes were measured as being attributed to alcohol use 

Thank you for this suggestion. All the adverse events are alcohol related, including crime, using data from turning point which collects data on alcohol related adverse events. The text was updated to make this clear on pages 9-10, line 195-198. 

See response to #3 above, also line 184 to 187 

5 

Perhaps an example could help non-expert readers follow? (e.g., a reduction in alcohol use might simultaneously reduce e.g: crime and hospital admissions, both of which could be considered separately 

We appreciate this comment. We’ve included an example as below: 

Page 10, line 201 to 204 (e.g., when the same patient utilises the ambulance, ED, and in-hospital care for a hospital admission, these events were considered as three separate patients and then the join effect was applied to attribute those events to that one patient) 

6 

this approach may be limited in its ability to handle complex variables and interactions. not sure if this standard in the industry- please clarify 

other confounding factors are discounted in my opinion (e.g., socioeconomic status, mental health conditions). Also, the model does not account for varying levels of alcohol consumption reduction. For example, small reductions in alcohol consumption may not result in a proportional decrease in adverse events, especially for those already engaging in harmful drinking patterns. Whereas, large reductions in consumption may have a more substantial impact, but the model does not differentiate between these effects - if you have the data for this, please include it in the analysis, if not please address this in the discussion 

We understand your concern regarding confounding. In economic evaluation and modelling, it is normal practice to use Excel or Triage for health economic modelling, without controlling for confounders. 

In this case, we’ve used the reduction in alcohol consumption to model averted alcohol related adverse events for risky alcohol consumers (by age group, gender and community). We acknowledge that the reduction may not be proportionate within this group and have indicated this as a limitation (please see response # 1). 

7 

why? This could lead to an incomplete assessment of the intervention's full value, as improvements in mental health, stress reduction, or quality of life are often central to public health initiatives and in the overall prevention of relapses in any alcohol related harms: Social Determinants of Health and Alcohol Use: 

Galea, S., Nandi, A., & Vlahov, D. (2004). The social epidemiology of substance use. Epidemiologic Reviews, 26(1), 36-52. 

If you have the data for this, please consider including it in the analysis, if not please address this limitation in the discussion 

We agree that this leads to an incomplete assessment. However, the CTC study didn’t collect quality of life data. We therefore applied a limited-social perspective. The method section has been updated to indicate this. 

We say limited-social perspective (on page 5) because it does not account for quality of life and wellbeing. 

This has also been considered in the limitation section (line 396). 

8 

I am wondering if the cost per event could vary by region? - please add a footnote to the table acknowledging this 

Thank you for this query. Cost per event was comparable and not different across regions. 

9 

could this capture complex interactions between multiple variables. For example, changes in youth exposure to the intervention may not have the same proportional impact on different outcomes ? 

We do not agree that bootstrapping in this study would adequately capture complex interactions between multiple variables. Typically, bootstrapping captures synchronous variations among all the model inputs. 

10 

I may not have understood this, but how was this conclusion derived if the BC analysis was only applied during the intervention period 

Not clear how the long-term impact has been measured in this study 

Yes, we agree – there is some misunderstanding: 

We were exploring the existing evidence in the literature in the discussion. We compared our results with other studies which included both short-term and long-term impact. We have only considered short-term impact in this study, as mentioned in the method and discussion. 

11 

Didn’t the primary analysis only consider the 1st cycle? or did the analysis also include other cycles but did not include in the ms? 

We appreciate this comment. In our analysis, we used the first cycle since all communities had the one cycle for comparability across the four communities. We acknowledge that communities can have more than one cycle and discover economies of scale, and therefore discussion about it was noteworthy. 

In the method section, we have now highlighted that ¾ of the communities had only one cycle (hence, only one had two cycles). 

Please see the response to comment #2. 

12 

I think some more limitations should be included both the Toumbourou study and the BCA rely on self-reported data to estimate alcohol use, which could lead to response bias. Self-reported measures may under-report actual consumption levels, leading to underestimated benefits of the intervention 

Thank for highlighting this. We have included this as a limitation as suggested. 

This change can be found on page 23, Line 362-365 

Self-reported data on alcohol use is a limitation due to the potential for response bias. Since adolescents are generally not expected to consume alcohol, these self-reported measures may under-report actual consumption levels, thereby obscuring the true benefits of the intervention. 

13 

I also wonder if the inclusion of a Cost effectiveness analysis might be suggested here? perhaps as future recommendation to avoid complicating this paper? this could give a bigger picture on the overall effectiveness of CTA when compared to alternative programs without necessarily quantifying it in monetary value… 

IF you do have data to do a cost effectiveness analysis, please consider adding this, if not, as suggested above, consider adding to recommendation 

Thank you for this suggestion. We included this in the conclusion as below (….incorporate more robust experimental designs,…). This has been modified as follows. 

We have revised this on Page 24 to 25, line 407-411 

Future research should examine the economic benefits of prevented youth tobacco and cannabis use as well as antisocial behavior as demonstrated in the Australian CTC Trial 6. Additionally, it should utilise more robust experimental designs, including randomised control trials, and consider average absolute reductions that accounts for age, gender, length of intervention period, and long-term benefits for cost effectiveness analysis. 

We believe that the revisions have strengthened our manuscript, and we hope it meets your expectations. Thank you once again for your insightful comments. 

Sincerely, 

Julie Abimanyi-Ochom 

Deakin Health Economics, Deakin University

---

## [Decision Letter · Decision Letter 2]

6 Nov 2024

Australian Communities That Care (CTC) intervention: Benefit-cost analysis of a community-based youth alcohol prevention initiative

PONE-D-24-08327R2

Dear Dr. Abimanyi-Ochom,

We’re pleased to inform you that your manuscript has been judged scientifically suitable for publication and will be formally accepted for publication once it meets all outstanding technical requirements.

Kind regards,

Daryl Higgins, PhD

Academic Editor

PLOS ONE

Additional Editor Comments (optional):

Reviewers' comments:

Reviewer's Responses to Questions

**Comments to the Author**

1. If the authors have adequately addressed your comments raised in a previous round of review and you feel that this manuscript is now acceptable for publication, you may indicate that here to bypass the “Comments to the Author” section, enter your conflict of interest statement in the “Confidential to Editor” section, and submit your "Accept" recommendation.

Reviewer #2: All comments have been addressed

2. Is the manuscript technically sound, and do the data support the conclusions?

Reviewer #2: Yes

3. Has the statistical analysis been performed appropriately and rigorously? 

Reviewer #2: Yes

4. Have the authors made all data underlying the findings in their manuscript fully available?

Reviewer #2: Yes

5. Is the manuscript presented in an intelligible fashion and written in standard English?

Reviewer #2: Yes

6. Review Comments to the Author

Reviewer #2: Thank you for addressing all the comments and for providing more clarity. There are no further comments from my end.

7. PLOS authors have the option to publish the peer review history of their article (what does this mean?). If published, this will include your full peer review and any attached files.

Reviewer #2: **Yes: **Dhatsayini Rattambige

---

## [Editor Report · Acceptance letter]

15 Nov 2024

PONE-D-24-08327R2 

PLOS ONE

Dear Dr. Abimanyi-Ochom, 

I'm pleased to inform you that your manuscript has been deemed suitable for publication in PLOS ONE. Congratulations! Your manuscript is now being handed over to our production team.

Kind regards, 

on behalf of

Professor Daryl Higgins 

Academic Editor

PLOS ONE